# A Novel BoHV-1-Vectored Subunit RVFV Vaccine Induces a Robust Humoral and Cell-Mediated Immune Response Against Rift Valley Fever in Sheep

**DOI:** 10.3390/v17030304

**Published:** 2025-02-23

**Authors:** Selvaraj Pavulraj, Rhett W. Stout, Shafiqul I. Chowdhury

**Affiliations:** Department of Pathobiological Sciences, School of Veterinary Medicine, Louisiana State University, Baton Rouge, LA 70803, USA; pavulraj.vet@gmail.com (S.P.); rstout1@lsu.edu (R.W.S.)

**Keywords:** BoHV-1, BoHV-1 vector, subunit RVFV vaccine, RVFV, immunogenicity, Gn and Gc, sheep, vaccine

## Abstract

Rift Valley fever (RVF) is a vector-borne zoonotic viral disease that causes abortion storms, fetal malformations, and neonatal mortality in livestock ruminants. In humans, RVF can lead to hemorrhagic fever, encephalitis, retinitis, or blindness, and about 1% of patients die. Since there are no registered vaccines for human use, developing RVF vaccines for use in animals is crucial to protect animals and prevent the spread of the virus from infecting humans. We recently developed a live bovine herpesvirus type 1 quadruple gene-mutant vector (BoHV-1qmv) that lacks virulence and immunosuppressive properties. Further, we engineered a BoHV-1qmv-vectored subunit Rift Valley fever virus (RVFV) vaccine (BoHV-1qmv Sub-RVFV) for cattle, in which a chimeric polyprotein coding for the RVFV Gc, Gn, and bovine granulocyte–macrophage colony-stimulating factor (GMCSF) proteins is fused but cleaved proteolytically in infected cells into individual membrane-anchored Gc and secreted Gn-GMCSF proteins. Calves vaccinated with the BoHV-1qmv Sub-RVFV vaccine generated moderate levels of RVFV-specific serum-neutralizing (SN) antibodies and cellular immune responses. In the current study, we repurposed the BoHV-1qmv Sub-RVFV for sheep by replacing the RVFV Gc and Gn ORF sequences codon-optimized for bovines with the corresponding ovine-codon-optimized sequences and by fusing the sheep GM-CSF ORF sequences with the Gn ORF sequence. A combined primary intranasal-plus-subcutaneous primary immunization induced a moderate level of BoHV-1 (vector)- and vaccine strain MP12-specific SN antibodies and MP-12-specific cellular immune responses. Notably, an intranasal booster vaccination after 29 days triggered a rapid (within 7 days) rise in MP-12-specific SN antibody titers. Therefore, the BoHV-1qmv-vectored subunit RVFV vaccine is safe and highly immunogenic in sheep and can potentially be an efficient subunit vaccine for sheep against RVFV.

## 1. Introduction

Rift Valley fever virus (RVFV) is an emerging pathogen that maintains high biodefense priority based on its threat to livestock because of its ability to cause human hemorrhagic fever and its potential for aerosol spread [1,2]. The primary hosts of RVFV are cattle, sheep, and goats. In livestock, Rift Valley fever virus (RVFV) outbreaks are notable for fatalities in younger animals and “abortion storms” among pregnant dams [3,4], which are the source of human infections. RVFV has been widely distributed in sub-Saharan Africa, with epizootic activity affecting animals in Kenya, Tanzania, Zambia, and Uganda. Rapid intercontinental commerce and a lack of effective control measures threaten to expand the geographic range of RVFV. A recent example is an expansion to the Arabian Peninsula [5,6,7].

Livestock are highly susceptible to RVFV infection and provide two critical ecological links between the primary vector, the *Aedes* sp. the floodwater mosquito, and the human population. First, livestock infected by the bite from transovarially infected *Aedes* sp. mosquitoes rapidly develop high viremias, allowing for the spillover of RVFV into secondary vectors (*Culex* sp. and *Anopheles* sp. mosquitoes) that are more likely to feed on humans [3]. Second, the high viral loads found in livestock are also a significant risk factor for human infection by direct contact with contaminated blood, tissues, and aborted fetal materials [8]. Veterinarians, farm workers, and other health personnel are at high risk of infection from direct contact with infected animals. Many historical outbreaks of RVFV livestock infection in Africa were initially detected when these workers came to human clinics.

The enveloped RVFV encodes a negative-sense single-stranded segmented RNA genome (S-, M-, and L-segments) 11.9 kilobase pairs (kbp) in size. The L-segment codes for viral polymerase protein (6404 nucleotides (nt); 2092 amino acids (aa)); the M-segment codes for two glycoproteins, i.e., amino-terminal glycoprotein (Gn) and carboxy-terminal glycoprotein (Gc), and two nonstructural proteins, i.e., NSm1 and NSm2 (3885 nt; 1197 aa) in a single open reading frame but cleaved proteolytically into individual proteins in the endoplasmic reticulum; and the S-segment (1690 nt) encodes for the viral nucleocapsid protein (N; 245 aa) and nonstructural protein NSs (265 aa). The Gn and Gc proteins form the heterodimer complex required for transport and maturation in the Golgi and Gn-Gc envelope incorporation [9]. RVFV Gn-Gc is the primary target of RVFV-specific neutralizing antibodies and the CD4-positive memory T-cell-mediated immune (CMI) response [10,11].

A robust and effective vaccination campaign targeting livestock in the affected regions has the direct dual benefit of protecting the lives of millions of animals and eliminating the risk of severe and sometimes lethal human RVF disease. Consequently, RVFV is uniquely suited for a one-health approach to prevent livestock and human diseases through animal vaccination. In essence, blocking the virus amplification step in livestock by vaccination could prevent RVFV outbreaks by reducing the potential for secondary vector spillover and eliminating the threat posed by infected livestock tissues [3].

The RVFV modified live virus (MLV) vaccine strain MP-12 was developed from the virulent ZH-548 strain. However, it retained residual virulence and caused abortion in ewes and teratogenic effects in newborn lambs. Additionally, in field conditions, there is a risk of the MP-12 strain reassorting with the wild type and potentially regaining virulence [12,13,14,15,16]. Experimentally, replication-competent Capripox viral vectors, particularly the lumpy skin disease virus (LSDV)-vectored subunit RVFV vaccine, are efficacious against LSDV and RVFV. However, they lack a serological marker to distinguish the vaccine strains from the circulating wild-type LSDV viruses. Over the years, there has been an unprecedented spread of LSDV from Africa into Europe, Russia, and Asia, and, importantly, there has been the emergence of novel recombinant LSDVs. Presumably, these events resulted from possible recombination between the LSDV MLV and gene-deleted LSDV vaccine virus with the circulating wild-type LSDV, goatpox virus (GTPV), and sheeppox virus (SHPV) [17]. This is probably why the LSDV-vectored RVFV subunit vaccine has not yet been licensed [18,19]. Similarly, replication-defective adenovirus- and modified vaccinia virus Ankara (MVA)-vectored subunit RVFV vaccines are also efficacious under experimental conditions. However, as in the case of LSDV-vectored vaccines, none are licensed or commercialized yet for RVFV in nonendemic or endemic countries [20].

We recently constructed a live bovine herpesvirus type 1 (BoHV-1) quadruple gene-mutated virus (BoHV-1qmv)-vectored subunit vaccine against RVFV (BoHV-1qmv Sub-RVFV). We demonstrated that calves immunized with the vaccine, both intranasally and subcutaneously only once, exhibited protective levels of RVFV-specific serum-neutralizing antibody titers and cell-mediated immune response (doi:10.3390/v15112183). Further, we showed that the BoHV-1qmv Sub-RVFV established latency in the trigeminal ganglionic (TG) neurons following primary intranasal vaccination. However, upon dexamethasone-induced latency reactivation, the vaccine virus was not transported to and shed in the nasal mucosa [21]. These results validated the safety and efficacy of the BoHV-1qmv-vectored subunit RVFV vaccine in calves [22].

In the current study, we repurposed the BoHV-1qmv Sub-RVFV vaccine by replacing its chimeric RVFV Gn-GMCSF (granulocyte–macrophage colony-stimulating factor) and Gc subunit proteins codon-optimized for bovines with the codons optimized for ovines (Sub-RVFV) and tested its safety and protective RVFV-specific immunogenicity. The results presented in this report revealed that the repurposed BoHV-1qmv Sub-RVFV is safe and induced moderate RVFV-specific serum-neutralizing antibody titers and RVFV-specific interferon-gamma (IFN-γ) secreting peripheral blood mononuclear cells (PBMCs) after combined primary intranasal–subcutaneous vaccination. Notably, subsequent intranasal booster vaccination induced an 8-fold (significant) rise in recall RVFV-specific neutralizing antibody response.

## 2. Materials and Methods

### 2.1. Cells and Medium

Madin–Darby ovine kidney (MDOK, #CRL-1633TM) cells were purchased from the American Type Culture Collection (ATCC^®®^, Manassas, VA, USA). The MDOK, Madin–Darby bovine kidney (MDBK) and KOP-R cells were propagated and maintained in Dulbecco’s modified Eagles medium (DMEM) (#10-017-CV; Corning^®^, Corning, NY, USA) supplemented with 10% heat-inactivated EquaFETAL serum (Atlas Biologicals, Fort Collins, CO, USA) and 1× antibiotic–antimycotic solution (#30-004-CI; Corning^®^). The PBMCs collected from the sheep for the cell-mediated immune response assay were cultured in complete RPMI medium (#11875085; Gibco™, Waltham, MA, USA) supplemented with heat-inactivated 10% Equa-FETAL serum, 2-Mercaptoethanol (50 µM), L-glutamine (20 mM), HEPES (25 mM), and 1× antibiotic–antimycotic solution.

### 2.2. Viruses

The BoHV-1 wild-type (wt) Cooper strain (Colorado-1) was obtained from the ATCC (#VR-864TM; ATCC^®®^), and low-passage viral stocks were maintained at −80 °C. An attenuated BoHV-1qmv virus lacking the U_L_49.5, gG, gE-CT, and US9 genes reported earlier [23] was used for the construction of the subunit RVFV vaccine virus (BoHV-1qmv Sub-RVFV) [22]. BoHV-1 recombinant vector and vaccine viruses for the sheep experiment were propagated in MDOK cells. The BoHV-1 wt and BoHV-1 recombinant viruses were titrated by plaque assay in MDBK cells [23]. The attenuated RVFV vaccine strain, MP-12, was obtained from Dr. Chris Mores (George Washington University, Washington, DC, USA), and low-passage virus stocks were propagated and titrated in MDBK cells.

### 2.3. Antibodies

RVFV anti-Gn- and anti-Gc-specific rabbit polyclonal peptide antibodies were synthesized commercially and characterized [22]. Mouse anti-V5 mAb (#R960-25; Thermo Fisher Scientific^®®^, Waltham, MA, USA) and anti-FLAG mouse mAb (#F1804; Sigma-Aldrich^®®^, St. Louis, MO, USA) were purchased. Donkey anti-mouse IgG Alexa Fluor 488 (#A-21202), goat anti-mouse IgG horseradish peroxidase (HRP) (#32430), and donkey anti-rabbit HRP conjugate (#31458) were purchased from Thermo Fisher scientific^®®^.

### 2.4. Construction of BoHV-1qmv Vector Virus Expressing the Chimeric RVFV Gn-GMCSF (Ovine) Fusion and Gc Proteins

The strategy and construction of the BoHV-1qmv vaccine vector and BoHV-1qmv Sub-RVFV vaccine virus for cattle expressing chimeric RVFV Gn-GMCSF (bovine)-Gc as polyprotein were reported previously [22,23]. To construct the BoHV-1qmv-vectored subunit RVFV vaccine virus for sheep, we first designed a chimeric subunit RVFV Gn–GMCSF-Gc gene cassette, as in [22,23], and the polyprotein coding chimeric RVFV Gn-GMCSF-Flag-P2A-Gc-V5 fused ORFs were codon-optimized for sheep (*Ovis aries*). The sheep GMCSF- ORF sequence (GenBank accession #NP_001009805.1) was fused with the Gn ORF (Figure 1, Appendix A). The 4.459 kb KpnI-HindIII fragment containing the entire chimeric RVFV Gn-GMCSF-Flag-P2A-Gc-V5 gene with the cytomegalovirus immediate–early (CMV IE) promoter and SV40 PolyA sequences was synthesized (Biomatik, Kitchener, ON, Canada) and delivered as a pUC57 clone (Figure 1. We incorporated the 4.459 kb fragment in the KpnI-HindIII-digested pgGΔ plasmid. The resulting pRVFV Gn-GMCSF-Gc insertion plasmid DNA was used for co-transfection with full-length BoHV-1qmv genomic DNA (Figure 1B) to generate a BoHV-1qmv Sub-RVFV-O recombinant virus as described before [22,23]. Several putative BoHV-1qmv Sub-RVFVs were plaque-purified, as confirmed by nucleotide sequencing and immunoblotting (described below). One selected BoHV-1qmv Sub-RVFV-O recombinant was selected for further studies.

### 2.5. Growth Kinetics and Plaque Size Assay

The growth kinetics of BoHV-1qmv Sub-RVFV-O were evaluated and compared with that of the BoHV-1 wt by standard one-step growth kinetics assay, as described previously (Vaccines 2021, 9, 46). To determine the cell-to-cell spread property of BoHV-1qmv Sub-RVFV-O compared with that of BoHV-1 wt, the average plaque sizes of wt and BoHV-1qmv Sub-RVFV viruses were determined by measuring approximately 150 randomly selected plaques of each virus under an inverted fluorescent microscope (Olympus IX71; Shinjuku City, Tokyo, Japan). As previously described, the actual plaque diameters were measured using ImageJ^®®^ software version 1.53t (National Institute of Health, Bethesda, MD, USA) [23].

### 2.6. Characterization of RVFV Gn and Gc Proteins

Western blot analysis was performed as before [22,23] to confirm the expression of chimeric RVFV Gn-GMCSF and Gc proteins by BoHV-1qmv Sub-RVFV in the infected KOP-R cells. The KOP-R cells were infected with BoHV-1qmv Sub-RVFV or RVFV vaccine strain MP12 at a multiplicity of infection (MOI) of 5. When the cytopathic effects (CPEs) were 80–90% (48 hrs post-infection for BoHV-1qmv Sub-RVFV and 72 hrs post-infection for MP-12), cells were harvested and processed for cell lysates as described previously [22]. The solubilized proteins were aliquoted and stored at −80 °C. SDS-PAGE and Western immunoblotting analysis were performed to validate the chimeric Gn-GMCSF and Gc protein expression by the BoHV-1qmv Sub-RVFV. MP-12-infected KOP-R cell lysates served as positive controls for Gn and Gc.

### 2.7. Animal Experiment

Animal infection, handling, sample collection, and euthanasia protocols were approved by the Louisiana State University Institutional Animal Care and Use Committee (Protocol #20-028). Fifteen four-month-old nonvaccinated Louisiana Native sheep breed (Gulf Coast sheep) lambs of either sex were obtained. The sheep were pre-tested for BoHV–1 serum-neutralizing (SN) antibody titers to ensure BoHV-1-free status. Sheep with <4 BoHV-1-specific SN antibody titers were selected for the study. The sheep were randomly divided into three groups, with five lambs in each group. Sheep in the individual groups were housed in pens well-separated from each other (by more than 100 feet) and located in the School of Veterinary Medicine’s large animal isolation barn. Foot baths were placed at the main entrance and in front of each pen.

Primary and booster immunization and sample collection schemes are shown in Figure 2. After a week of acclimatization, the vector control group was given, intranasally (IN), BoHV-1qmv vector 5 × 10^7^ PFU/nostril (a total of 1 × 10^8^ PFUs IN) and, subcutaneously (SQ), 1 × 10^7^ PFU. Similarly, five sheep in each of the prime and prime–boost groups were given the BoHV-1qmv Sub-RVFV vaccine, IN, 5 × 10^7^ PFU/nostril (a total of 1 × 10^8^ PFUs IN), and, SQ, 1 × 10^7^ PFU. On 29 days post-vaccination (dpv), the five sheep in the prime–boost vaccine group received a booster immunization intranasally, as in the case of primary IN vaccination (a total of 1 × 10^8^ PFUs) with the BoHV-1qmv Sub-RVFV vaccine. Blood for serum and PBMCs and nasal swabs were collected at 0, 3, 6, 7, 14, 21, 29, 32, 36, and 45 dpv (Figure 2). The experiment was terminated at 45 dpi, and sheep were euthanized with Euthasol^®®^ (Euthanasia Solution; pentobarbital sodium and phenytoin sodium).

### 2.8. Clinical Examination, Sample Collection, and Processing

Sheep were clinically assessed for rectal temperature, feed and water intake, and nasal and ocular discharge from 0 to 45 dpv (until euthanasia). The EDTA blood and nasal swab samples were collected as shown in Figure 2 and processed for PBMCs, serum collection, and virus isolation. Nasal swabs were collected in 1 mL of cell culture medium supplemented with 2× antibiotic–antimycotic solution. Vaccine virus/vector nasal shedding was determined by plaque assay as well as qPCR analysis [22,23].

### 2.9. Serum Virus Neutralization Assay for BoHV-1qmv- and BoHV-1qmv Sub-RVFV-Group Sheep Against RVFV Vaccine Strain MP-12 and BoHV-1 by Plaque Reduction

Heat-inactivated sheep serum samples (56 °C for 30 min) were used for the BoHV-1- and RVFV-specific plaque reduction assays. The BoHV-1- and RVFV-specific plaque reduction assays were described previously [22].

### 2.10. DNA Isolation and Quantitative PCR (qPCR)

To quantify the BoHV-1qmv vector and BoHV-1qmv Sub-RVFV genome copies in the nasal swabs following the primary and booster vaccinations, total DNA was isolated using the QIAamp^®®^ DNA mini kit (#51306; Qiagen, Hilden, North Rhine–Westphalia, Germany). BoHV-1 genome copies were determined by TaqMan probe-based real-time qPCR in the ABI PRISM™ 7900HT Sequence Detection System (Applied Biosystems, Waltham, MA, USA), targeting the major capsid protein (VP5) ORF coding sequence using primer pairs described previously [22]. Each time, the PCR reaction setup was run with six standards of known quantity (10^1^ to 10^6^ copies per reaction). BoHV-1 genome copies in the nasal samples (normalized to 100 ng of total DNA) were compared with the generated standard curves. The assay was duplicated, and results were expressed as BoHV-1 genome copies per 100 ng of DNA.

### 2.11. RVFV-Specific Cell-Mediated Immune Response Assay

Aliquots of isolated PBMCs collected from the immunized sheep on days 0, 14, and 21 post-vaccination and stored in liquid nitrogen were thawed and seeded at a rate of 0.5 × 10^6^ cells/well in a 96-well plate (in 100 µL volume). After overnight incubation in RPMI medium at 37 °C, PBMCs were stimulated with 10 µg/mL of heat-inactivated RVFV antigens as described previously [22].

### 2.12. Statistical Analysis

Statistical analyses were performed using GraphPad PRISM^®®^ 5.01 software (San Diego, CA, USA). Normally distributed group samples were analyzed with a two–way ANOVA followed by Bonferroni post-tests to compare replicate means by row. A ‘*p*’ value of less than 0.05 was considered significant for all analyses.

## 3. Results

### 3.1. The Recombinant BoHV-1qmv Sub-RVFV-O Vaccine Virus Expresses the Chimeric RVFV Gn-GMCSF-FLAG and Gc-V5 Self-Cleavable Polyproteins

Sequence analyses of the constructed BoHV-1qmv Sub-RVFV-O vaccine virus genomic region spanning the chimeric gene, expressing RVFV Gn-GMCSF (ovine)-P2A, and Gc ORF sequences (Appendix A) and its BoHV-1-specific flanking sequences (Figure 1; approx. 1000 bp on each side) validated the integrity and their appropriate insertion at the gG deletion locus. Further, the expression of proteolytically cleaved (P2A), chimeric FLAG-tagged Gn-GMCSF and V5-tagged Gc proteins in the BoHV-1qmv Sub-RVFV-infected KOP-R cell lysates was verified in comparison to the MP-12 expressed Gn and Gc proteins by SDS–PAGE/Western immunoblotting. As depicted in Figure 3, both the RVFV Gn-specific and FLAG-specific antibodies detected a protein band with a molecular mass of approximately 75 kDa in the case of BoHV-1qmv Sub-RVFV-infected cell lysates (Figure 3A,C). Similarly, the RVFV Gc-specific antibody detected approximately a 61 kDa band in the RVFV MP-12-infected KOP-R lysate (Figure 3B,D).

The estimated molecular mass (https://web.expasy.org/compute_pi/; accessed on 12 December 2022) of the chimeric FLAG-tagged Gn-GMCSF protein is 70.1 kDa (RVFV Gn—54.8 kDa; ovine GMCSF—14.3 kDa; FLAG tag—1 kDa), and the molecular mass of MP-12 Gn is 58.8 kDa, whereas the estimated molecular mass of the RVFV Gc-V5 chimeric protein is 56.8 kDa (55.44 kDa for Gc and 1.4 kDa for V5). The molecular mass increases by approximately 5 kDa in the cases of Gn-GMCSF-FLAG and Gc-V5, which is consistent with the predicted glycosylation sites, as reported earlier [22].

### 3.2. Like the BoHV-1 wt, the BoHV-1qmv Sub-RVFV Vaccine Virus Replicated with a Similar Kinetics and Virus Yield in MDBK Cells but Produced Smaller Plaques

Three independent assays were performed to determine the plaque sizes and one-step growth kinetics of the BoHV-1qmv Sub-RVFV virus relative to the BoHV-1 wt virus. The results presented in Figure 4 show that the BoHV-1qmv Sub-RVFV produced significantly smaller plaques (about a 72% reduction in plaque size) than the BoHV-1 wt virus. However, the one-step growth kinetics and virus yield of the BoHV-1qmv Sub-RVFV virus were identical to those of the BoHV-1 wt (Figure 4).

### 3.3. The BoHV-1qmv Sub-RVFV-O Vaccine Virus Is Highly Attenuated and Safe in Immunized Sheep

Following the combined intranasal (IN) and subcutaneous (Subcut) primary immunization, the sheep remained clinically healthy regardless of whether they were vaccinated with BoHV-1qmv Sub-RVFV-O or the parental BoHV-1qmv vector virus (Appendix A).

### 3.4. Nasal Virus Shedding Following Primary and Booster Immunization

Following intranasal immunization, both BoHV-1qmv (vector control) and BoHV-1qmv Sub-RVFV (prototype vaccine) replicated in the nasal mucosae of the sheep, as evidenced by BoHV-1-specific qPCR and viral plaque assays (Figure 5 and Figure 6 and Appendix A). Regardless of the vector control or BoHV-1qmv Sub-RVFV vaccine, on 3, 6, and 7 dpv, most sheep shed the virus in nasal secretions, according to qPCR results (Figure 5 and Appendix A). Further, BoHV-1qmv Sub-RVFV could be isolated in cell cultures from three and two sheep’s nasal swabs on 3 and 6 dpv, respectively (Figure 6 and Appendix A). As expected, by 14 dpv, both vaccine viruses established latency, and there was no trace of the virus in the nasal swabs of animals in either of the vaccinated groups (Figure 5, Appendix A).

On 29 dpv, five sheep in the BoHV-1qmv Sub-RVFV-vaccinated group received intranasal booster vaccination with the same dose as the primary immunization. Mean BoHV-1-specific genome copy values of 1.1 × 10^3^ and 3.2 × 10^3^ were detected in nasal swabs collected on 3 and 6 days post-booster (dpb), respectively (Figure 5 and Appendix A). Also, on 3 dpb, BoHV-1qmv Sub-RVFV was isolated by plaque assay from the nasal swabs of two of the five sheep that received booster vaccinations (Figure 5 and Appendix A).

### 3.5. BoHV-1-Vector-Specific and RVFV-Specific SN Antibody Titers in Sheep in Response to Primary (BoHV-1qmv and BoHV-1qmv Sub-RVFV) and Booster (BoHV-1qmv Sub-RVFV) Vaccinations

On the day of primary vaccination (0 dpv), all the sheep (ten vaccinated with BoHV-1qmv Sub-RVFV and five vaccinated with BoHV-1qmv) were negative for BoHV-1- and RVFV (MP-12)-specific serum-neutralizing antibodies. However, moderately high BoHV-1-specific (vector-specific) neutralizing antibodies (mean SN titers of 25) were detected by seven dpv, in both BoHV-1 qmv- and BoHV-1 qmv Sub-RVFV-vaccinated sheep. Significantly, by 21 days post-primary vaccination, BoHV-1-specific serum-neutralizing antibody titers in qmv vector- and qmv Sub-RVFV-vaccinated sheep rose to 65 and 62 (>60 folds), respectively, which values were statistically significant (*p* < 0.001) and exceeded the threshold seroconversion level (>4-fold rise in SN titers) by 15 folds (Figure 7A, Appendix A).

At 29 days post-vaccination (dpv), five of the ten sheep received the BoHV-1qmv Sub-RVFV vaccine (Figure 2). Their mean BoHV-1-specific SN titers on that day, or zero-day post-booster (0 dpb), was 19 (Figure 7A, Appendix A). However, 7 and 16 days after the booster (7 dpb and 16 dpb), the mean BoHV-1-specific SN antibody titers in the qmv Sub-RVFV prime–boost group were 141 and 101, respectively, which were statistically highly significant (*p* < 0.001) because they represented an increase of more than 7-fold compared to that on 0 dpb. Therefore, there was no apparent interference of the pre-existing vector-specific neutralizing antibodies with the active memory immune response when the qmv Sub-RVFV-group sheep received the intranasal booster vaccination. As expected, the mean BoHV-1-specific SN titers in the qmv and qmv Sub-RVFV prime groups of sheep that did not receive the booster vaccination fell further: in the qmv prime group, from 21 (29 dpv) to 16 and 14 (on 36 and 45 dpv, respectively), and in the qmv Sub-RVFV-prime group, from 19 (29 dpv) to 13 on 36 and 45 dpv (Figure 7A, Appendix A).

The results in Figure 7B (Appendix A) also demonstrate that the primary BoHV-1qmv Sub-RVFV vaccination (combined IN and Subcut) induced mean (based on ten sheep) MP-12 (RVFV vaccine)-specific serum-neutralizing antibody titers of 8, 11, and 26 on 7, 14, and 21 dpv, respectively. Therefore, the qmv Sub-RVFV vaccination induced a significant level of anti-RVFV envelope glycoprotein Gc- and Gn-specific neutralizing antibody titers as early as 7 dpv (*p* < 0.05) and at 14 dpv (*p* < 0.01). In particular, the SN titer of 26, a 26-fold rise on 21 dpv compared with the value on 0 dpv, was highly significant (*p* < 0.001).

At 29 dpv (0 dpb), the RVFV-specific SN antibody titers in the qmv Sub-RVFV-Prime-group sheep dropped slightly (from 26 on 21 dpv to 17) when five of the ten primarily vaccinated sheep received the intranasal qmv Sub-RVFV booster vaccination (qmv Sub-RVFV Prime–Boost group). As shown in Figure 7, the BoHV-1qmv Sub-RVFV booster-vaccinated sheep (qmv Sub-RVFV-Prime–Boost) also showed more than a four-fold (4.4-fold) rise in memory RVFV (MP-12)-specific mean neutralizing antibody titers, from 17 to 75 (significant at *p* < 0.01), by seven days post-booster, which dropped slightly to 63 by 16 days post-booster. In contrast, the mean MP-12-specific SN antibody titers in the qmv Sub-RVFV-Prime sheep group (which received only single IN plus Subcut vaccination) and the corresponding MP-12-specific SN antibody titers on 36 and 45 dpv dropped from 17 (29 dpv) to 10 and 9, respectively. Therefore, following the post-BoHV-1qmv Sub-RVFV intranasal booster vaccination, the RVFV-specific SN antibody titers rose by 7.5- and 6.3-fold on 7 dpb and 16 dpb, respectively, compared to the pre-booster SN titers, which values were significant (*p* > 0.001). These results demonstrated that BoHV-1qmv and BoHV-1qmv Sub-RVFV vaccine vectors are highly immunogenic and that intranasal booster vaccinations induced significantly higher vaccine vector-specific memory-neutralizing antibody levels. Further, the results also demonstrated that pre-existing vector-specific immunity did not interfere with the booster intranasal vaccination.

### 3.6. RVFV-Specific Cell-Mediated Immune Response (CMI) in the BoHV-1qmv Sub-RVFV-Immunized Sheep

To determine the RVFV-specific cellular immune response in the BoHV-1qmv Sub-RVFV vaccinated sheep, we stimulated in vitro the PBMCs collected from the immunized sheep on 0, 14, and 21 dpv with heat-killed RVFV MP-12 vaccine strain antigens and measured the IFN–γ mRNA transcript expression (Figure 8, Appendix A).

The results revealed that PBMCs collected from the BoHV-1 qmv Sub-RVFV-vaccinated sheep on 14 dpv and 21 dpv following stimulation with the heat-killed MP-12 antigen exhibited 4.85-fold and 3.5-fold increases in RVFV-specific IFN–γ mRNA transcription, respectively, compared with the corresponding unstimulated PBMCs. As expected, PBMCs collected from all the sheep on the day of primary vaccination (0 dpv) were not stimulated by the heat-killed MP-12 antigen, and there was no increase in IFN-γ mRNA. Similarly, PBMCs collected on 14 dpv and 21 dpv from BoHV-1qmv-vaccinated sheep also were not stimulated with the heat-killed MP-12 antigen. These results demonstrated that the chimeric subunit Gn and Gc proteins expressed by qmv Sub-RVFV specifically induced a cellular immune response upon vaccination.

## 4. Discussion

RVFV can be devastating among livestock and lethal in humans, and it can have widespread economic effects. Consequently, RVFV is uniquely suited for a one-health approach to preventing livestock and human diseases through animal vaccination [24].

We recently developed a live BoHV-1qmv-vectored subunit RVFV (Sub-RVFV) vaccine and demonstrated its protective immunogenicity in cattle [22]. Further, we showed that, following intranasal inoculation, like the BoHV-1 wt, both the BoHV-1qmv (vaccine vector) and the BoHV-1qmv Subunit-RVFV vaccine virus established latency and were reactivated and replicated in TG neurons following dexamethasone treatments. But, unlike the BoHV-1 wt, they were not transported from the TG to the nasal epithelium and shed [21].

In this study, we repurposed the BoHV-1qmv Sub-RVFV for sheep by replacing the chimeric Gn-Gc (codon-optimized for bovines) and the bovine GM-CSF with the sheep-codon-optimized Gn-Gc and sheep GM-CSF, respectively. We demonstrated that combined primary intranasal and subcutaneous vaccination of sheep induced moderately high BoHV-1- and RVFV-specific SN antibody titers. Notably, BoHV-1qmv Sub-RVFV-vaccinated sheep also generated RVFV-specific cellular immune responses. Significantly, the intranasal BoHV-1qmv Sub-RVFV booster vaccination induced a rapid memory immune response, resulting in very high BoHV-1- and RVFV-MP-12-specific SN antibody titers. These results demonstrated that pre-existing BoHV-1-specific SN antibodies, generated after the primary BoHV-1qmv Sub-RVFV combined intranasal–subcutaneous vaccination, did not interfere with the booster intranasal immunization.

Previously, we tested the protective immunogenicity of BoHV-1qmv Sub-RVFV (a prototype live subunit RVFV vaccine) in calves. Based on the results [22], the mean RVFV (MP12 strain)-specific SN antibody titers rose to approx. 6.46 after only 7 dpv. At 21 dpv, the MP12-specific SN antibody titer rose to 17 (a 17-fold increase compared to that on 0 dpv), and at euthanasia (33 dpv) the RVFV-specific mean SN titer dipped only slightly to 14. The threshold for neutralizing antibody titers correlating with protection is not yet known for cattle. However, in mice, RVFV SN antibody titers correlate with 75–100% protection based on survival ranges from 1:5 to 1:20 [25]. Based on the data from this study in sheep, we expect that the intranasal booster immunization of cattle with the cattle-specific BoHV-1qmv Sub-RVFV (codon-optimized for bovines) will also result in a rapid memory immune response, resulting in high vector- and RVFV-MP-12-specific neutralizing antibody titers in cattle. In the near future, we plan to validate whether the RVFV-specific neutralizing antibody titers in cattle, sheep, and goats following the combined primary intranasal–subcutaneous vaccination are adequate for protection against a virulent RVFV challenge or require the intranasal booster vaccination.

## 5. Conclusions

Our vaccine efficacy study revealed that sheep immunized with the live BoHV-1qmv Sub-RVFV - vaccine using a combined intranasal–subcutaneous and prime–boost regimen is safe and stimulated significantly high SN antibody titers against the subunit vaccine vector (BoHV-1) and RVFV-MP-12. The intranasal booster vaccination was not affected by pre-existing vector-specific SN antibody titers resulting from the primary combined intranasal–subcutaneous vaccination.

## Figures and Tables

**Figure 1 viruses-17-00304-f001:**
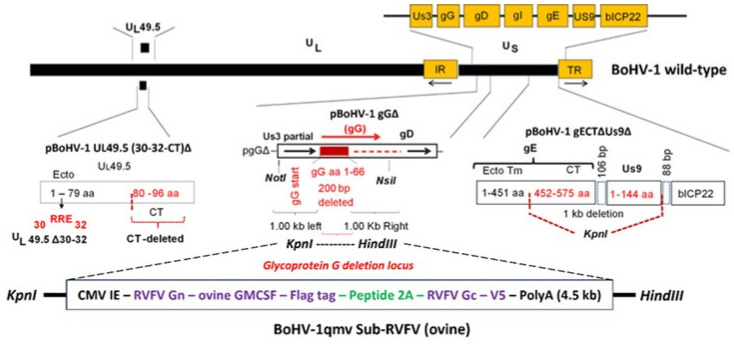
Schematic of bovine herpesvirus virus type 1 (BoHV-1) genome showing the UL49.5, glycoprotein G (gG), gE cytoplasmic tail (gE-CT), and Us9-deletion loci and the pBoHV-1 gECT∆-Us9∆, pBoHV-1 gG∆, and pBoHV-1 UL49.5(30-32-CT)∆ plasmids used to construct the BoHV-1qmv vector. The chimeric RVFV glycoprotein Gn–ovine GMCSF fusion and p2A and Gc sequences inserted into the gG-deletion locus of the BoHV-1qmv genome resulted in BoHV-1qmv Sub-RVFV-O.

**Figure 2 viruses-17-00304-f002:**
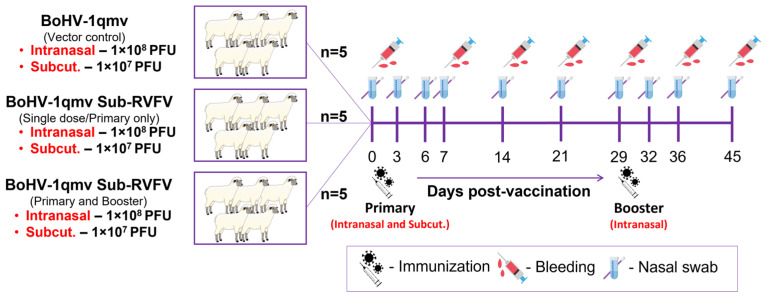
Schematic showing the sheep experiment’s immunization and sample collection scheme. Intranasal—intranasal inoculation; Subcut.—subcutaneous injection; PFU—plaque-forming units.

**Figure 3 viruses-17-00304-f003:**
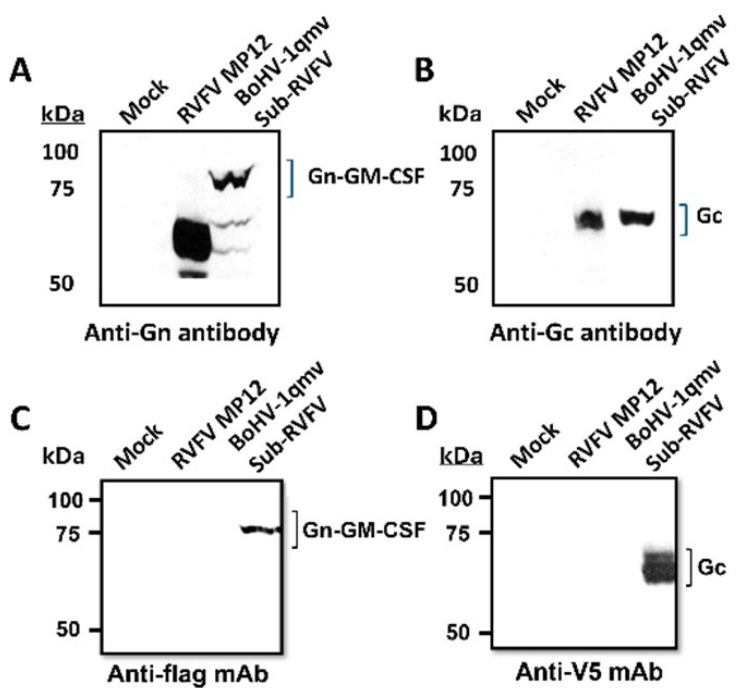
The RVFV Gn- and Gc-specific protein bands were absent in the mock-infected KOP-R cell lysates. Immunoblot analysis of the BoHV-1qmv Sub-RVFV expressing chimeric RVFV Gn-GMCSF and RVFV Gc proteins (codon-optimized for ovines) using rabbit anti-RVFV Gn (**A**), rabbit anti-RVFV Gc (**B**), anti-FLAG monoclonal antibodies (mAbs) (**C**), and anti-V5 mAbs (**D**). The RVFV Gn- and Gc-specific protein bands were absent in the mock-infected KOP-R cell lysates.

**Figure 4 viruses-17-00304-f004:**
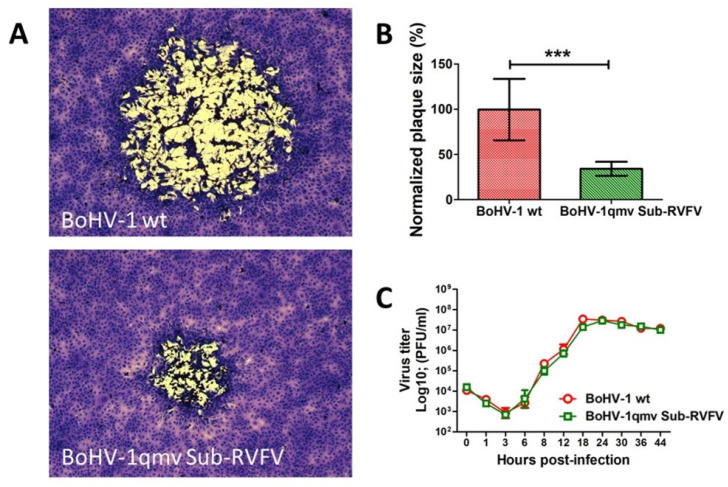
In vitro characterization of BoHV-1qmv Sub-RVFV-O vaccine virus. (**A**) Shown are the representative plaque morphologies of BoHV-1 wild-type and BoHV-1qmv Sub-RVFV-O viruses. (**B**) The bar graph shows the average plaque sizes of at least 50 plaques with SDs (*** *p* < 0.001). (**C**) One-step growth kinetics and virus yields of BoHV-1qmv Sub-RVFV-O at different times post-infection compared with those of BoHV-1 wild type.

**Figure 5 viruses-17-00304-f005:**
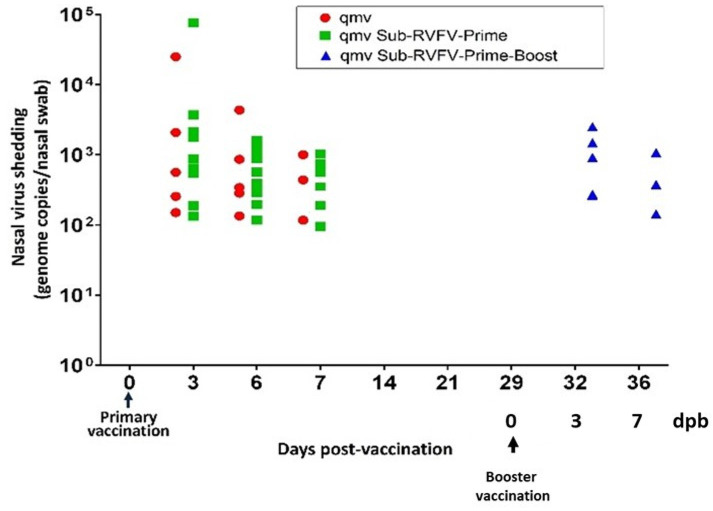
Nasal shedding of both BoHV-1qmv (qmv) vector and BoHV-1qmv Sub-RVFV (qmv Sub-RVFV) in immunized sheep determined by qPCR. Following immunization, DNA was isolated from nasal swabs, and BoHV-1-specific qPCR was performed. BoHV-1 genome copy numbers were calculated according to the CT values of a standard curve. Five sheep each were in the qmv (vector control) and qmv Sub-RVFV-Prime–Boost groups, and ten were in the qmv Sub-RVFV-Prime group. Shown are the copy numbers of the BoHV-1 genome per nasal swab. The dot-plot graph represents individual values in each group (Appendix A).

**Figure 6 viruses-17-00304-f006:**
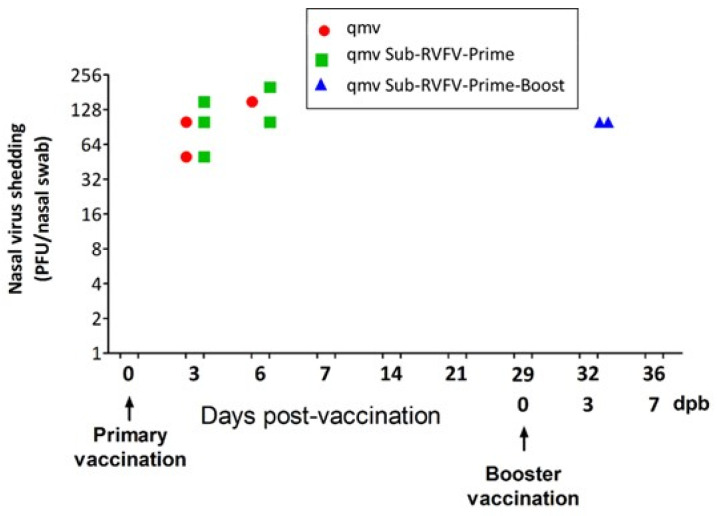
Nasal shedding of BoHV-1qmv vector (qmv)- and BoHV-1qmv Sub-RVFV (qmv Sub-RVFV)-immunized sheep assessed by virus isolation. Viruses isolated from animals’ nasal swabs after immunization with BoHV-1qmv vector or BoHV-1qmv Sub-RVFV vaccine (prime and prime–boost groups) were titrated in confluent KOP-R cells by plaque assay. The virus titers of each sheep (in plaque-forming units/nasal swab, PFU/nasal swab) are shown. The dot-plot graph represents individual values in each group (Appendix A).

**Figure 7 viruses-17-00304-f007:**
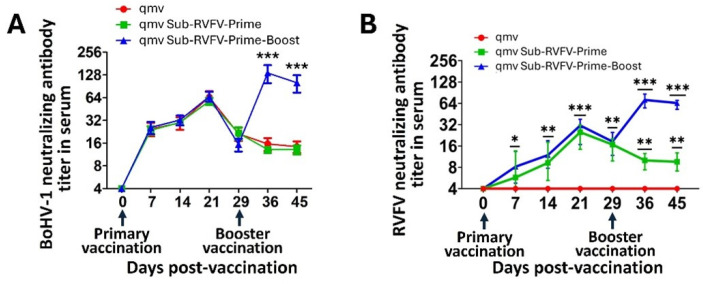
(**A**) Bovine herpesvirus type 1 (BoHV-1)- and (**B**) Rift Valley fever virus (RVFV)-specific serum-neutralizing antibody titers in sheep following primary BoHV-1qmv (qmv) and BoHV-1qmv Sub-RVFV (qmv Sub-RVFV-Prime) and booster qmv Sub-RVFV (qmv Sub-Prime–Boost) immunization. *—*p* < 0.05; **—*p* < 0.01; ***—*p* < 0.001.

**Figure 8 viruses-17-00304-f008:**
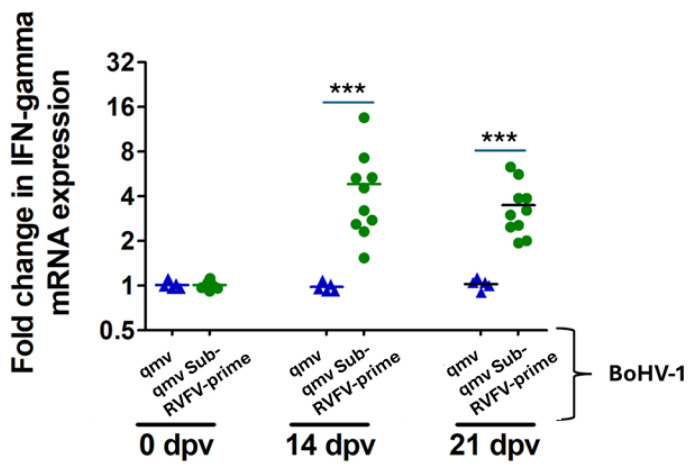
RVFV (MP12)-specific cellular immune response assay on PBMCs collected from BoHV-1qmv (vector control; blue triangles)- and BoHV-1qmv Sub-RVFV (green circles)-immunized sheep after primary vaccination. Data shown are fold changes in IFN-gamma mRNA expression levels after in vitro stimulation of cultured PBMCs with heat-killed MP-12 antigens, collected from sheep at 0, 14, and 21 dpv. Two-way ANOVA followed by Bonferroni post-tests to compare replicate means by row; *** *p* < 0.001.

## Data Availability

Data is contained within the article or Appendix A.

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
