# Peer review of "A Novel BoHV-1-Vectored Subunit RVFV Vaccine Induces a Robust Humoral and Cell-Mediated Immune Response Against Rift Valley Fever in Sheep"

_viruses, 2025, doi:10.3390/v17030304_

Round 1
Reviewer 1 Report
Comments and Suggestions for Authors
Overall, this is a comprehensive study on the development and evaluation of a novel BoHV-1-vectored subunit vaccine (BoHV-1qmv Sub-RVFV) against Rift Valley Fever Virus (RVFV) in sheep. The authors used a quadruple gene-mutant bovine herpesvirus type 1 (BoHV-1qmv) vector expressing RVFV envelope proteins Gn (fused with ovine GM-CSF) and Gc. The study found that intranasal and subcutaneous immunization induces robust humoral and cell-mediated immune responses in sheep, with intranasal booster vaccination significantly enhancing neutralizing antibody titers. The vaccine was shown to be safe, highly immunogenic, and capable of inducing memory responses without interference from pre-existing vector-specific antibodies.
While the data robustly support the conclusions, some weaknesses and confusing aspects in the presented data need addressing:
Comments:
1. The manuscript cites Pavulraj et al. multiple times (e.g., lines 109, 110, 138, 146, 156, 159, 169, 192, 198, 200, 235, 238) with conflicting publication years (2023 and 2024). However, the reference list includes only two unrelated Pavulraj et al. publications (2020 and 2022), neither of which related to RVFV. This discrepancy requires clarification.
References:
o Pavulraj, S., et al. (2022). Deleted Pseudorabies Virus-Vectored Subunit PCV2b and CSFV Vaccine Protects Pigs against PCV2b Challenge and Induces Serum Neutralizing Antibody Response against CSFV. Vaccines (Basel), 10, doi:10.3390/vaccines10020305.
o Pavulraj, S., et al. (2020). Equine Herpesvirus Type 1 Modulates Cytokine and Chemokine Profiles of Mononuclear Cells for Efficient Dissemination to Target Organs. Viruses, 12, 578. doi:10.3390/v12090999.
2. The study does not include RVFV challenge experiments. Critical questions remain unanswered: Does the neutralizing antibody induced by the vectored vaccine confer protection against RVFV? Is there a threshold for neutralizing antibody titers that correlates with protection? Additionally, how long do these antibodies persist? Long-term data on neutralizing antibody titers are needed to address these concerns.
Comments on the Quality of English Language
OK
Author Response
Comments and Response
Reviewer 1
Overall, this is a comprehensive study on the development and evaluation of a novel BoHV-1-vectored subunit vaccine (BoHV-1qmv Sub-RVFV) against Rift Valley Fever Virus (RVFV) in sheep. The authors used a quadruple gene-mutant bovine herpesvirus type 1 (BoHV-1qmv) vector expressing RVFV envelope proteins Gn (fused with ovine GM-CSF) and Gc. The study found that intranasal and subcutaneous immunization induces robust humoral and cell-mediated immune responses in sheep, with intranasal booster vaccination significantly enhancing neutralizing antibody titers. The vaccine was shown to be safe, highly immunogenic, and capable of inducing memory responses without interference from pre-existing vector-specific antibodies.
While the data robustly support the conclusions, some weaknesses and confusing aspects in the presented data need addressing:
- Comments: The manuscript cites Pavulraj et al. multiple times (e.g., lines 109, 110, 138, 146, 156, 159, 169, 192, 198, 200, 235, 238) with conflicting publication years (2023 and 2024). However, the reference list includes only two unrelated Pavulraj et al. publications (2020 and 2022), neither of which related to RVFV. This discrepancy requires clarification.
References: i) Pavulraj, S., et al. (2022). Deleted Pseudorabies Virus-Vectored Subunit PCV2b and CSFV Vaccine Protects Pigs against PCV2b Challenge and Induces Serum Neutralizing Antibody Response against CSFV. Vaccines (Basel), 10, doi:10.3390/vaccines10020305. ii) Pavulraj, S., et al. (2020). Equine Herpesvirus Type 1 Modulates Cytokine and Chemokine Profiles of Mononuclear Cells for Efficient Dissemination to Target Organs. Viruses, 12, 578. doi:10.3390/v12090999
Response: This was an error, and the references above have been replaced with the correct references (References # 21 and 22 are on the list).
- Comments: The study does not include RVFV challenge experiments. Critical questions remain unanswered: Does the neutralizing antibody induced by the vectored vaccine confer protection against RVFV? Is there a threshold for neutralizing antibody titers that correlates with protection? Additionally, how long do these antibodies persist? Long-term data on neutralizing antibody titers are needed to address these concerns.
Response: The USDA grant that funded the project did not include funding for the challenge experiments. We have a pending USDA grant in which we proposed the vaccination-challenge experiments. While the threshold level of anti-RVFV-specific neutralizing antibodies correlating to protection against challenge in cattle is unknown, it is known for mice. This is now discussed in the revised version.
Reviewer 2
In the manuscript by Pulvaraj et al, the authors re-purpose a previously developed RVFV vaccine (based on an attenuated bovine herpesvirus vector) for its use in sheep. The vector design strategy described is similar to that previously designed for use in cattle. The authors demonstrate that this vaccine can recombinantly express both RVFV glycoproteins, can induce both cell-mediated immunity, as well as generate neutralizing antibodies. The manuscript is well-written, and suitable for publication, provided that the following concerns are addressed:
Minor Concerns:
- Comment: The Abstract is unnecessarily long and should only summarize the study (i.e., there really shouldn't be references, nor should there be hard data in the Abstract). Please make it shorter
Response: We have shortened the abstract and removed the reference.
- Comment: In supplementary Figure 4, sheep 36, 41 and 45 are underlined? Is that supposed to be like that?
Response: It could be because of type setting. We could not see any underlining issues from our end.
- Comment: When discussing future considerations, perhaps the authors should mention that they should validate the vaccine against an actual RVFV experimental infection.
Response: It is noted as suggested.
Major Concerns:
- Comment: The authors should discuss why so many vaccinations are necessary. Is it really necessary to vaccinate both IN and SC the first time, followed by another IN as a boost? Even RVFV DNA vaccines (which have been disregarded a practical vaccine candidate) require only 2 SC vaccinations to get full protection. The authors should explain why their vaccine requires 3 administrations over two periods (as opposed to other vaccine candidates, which require less).
Response: The rationale for using simultaneous intranasal and subcutaneous routes for the primary vaccination was that intranasal immunization induces better mucosal and cellular immune responses, and subcutaneous immunization induces higher levels of antibodies. Intranasal booster immunization aimed to determine i) the memory immune response and ii) to validate that the preexisting antibodies do not interfere with the intranasal vaccination.
Reviewer 2 Report
Comments and Suggestions for Authors
In the manuscript by Pulvaraj et al, the authors re-purpose a previously developed RVFV vaccine (based on an attenuated bovine herpesvirus vector) for its use in sheep. The vector design strategy described is similar to that previously designed for use in cattle. The authors demonstrate that this vaccine can recombinantly express both RVFV glycoproteins, can induce both cell-mediated immunity, as well as generate neutralizing antibodies. The manuscript is well-written, and suitable for publication, provided that the following concerns are addressed:
Minor Concerns:
- The Abstract is unnecessarily long and should only summarize the study (i.e. there really shouldn't be references, nor should there be hard data in the Abstract). Please make it shorter.
-In supplementary Figure 4, sheep 36, 41 and 45 are underlined? Is that supposed to be like that?
-When discussing future considerations, perhaps the authors should mention that they should validate the vaccine against an actual RVFV experimental infection.
Major Concerns:
- The authors should discuss why so many vaccinations are necessary. Is it really necessary to vaccinate both IN and SC the first time, followed by another IN as a boost? Even RVFV DNA vaccines (which have been disregarded a practical vaccine candidate) require only 2 SC vaccinations to get full protection. The authors should explain why their vaccine requires 3 administrations over two periods (as opposed to other vaccine candidates which require less).
Author Response

(The authors gave the same response as above.)
